# Detecting Toe-Off Events Utilizing a Vision-Based Method

**DOI:** 10.3390/e21040329

**Published:** 2019-03-27

**Authors:** Yunqi Tang, Zhuorong Li, Huawei Tian, Jianwei Ding, Bingxian Lin

**Affiliations:** 1School of Forensic Science, People’s Public Security University of China, Beijing 100000, China; 2School of Criminal Investigation and Counter Terrorism, People’s Public Security University of China, Beijing 100000, China; 3School of Information Engineering and Network Security, People’s Public Security University of China, Beijing 100000, China; 4Jiangsu Center for Collaborative Innovation in Geographical Information Resource Development and Application, Nanjing 210000, China; 5Key Laboratory of Virtual Geographic Environment, Nanjing Normal University, Ministry of Education, Nanjing 210000, China

**Keywords:** toe-off detection, gait event, silhouettes difference, convolutional neural network

## Abstract

Detecting gait events from video data accurately would be a challenging problem. However, most detection methods for gait events are currently based on wearable sensors, which need high cooperation from users and power consumption restriction. This study presents a novel algorithm for achieving accurate detection of toe-off events using a single 2D vision camera without the cooperation of participants. First, a set of novel feature, namely consecutive silhouettes difference maps (CSD-maps), is proposed to represent gait pattern. A CSD-map can encode several consecutive pedestrian silhouettes extracted from video frames into a map. And different number of consecutive pedestrian silhouettes will result in different types of CSD-maps, which can provide significant features for toe-off events detection. Convolutional neural network is then employed to reduce feature dimensions and classify toe-off events. Experiments on a public database demonstrate that the proposed method achieves good detection accuracy.

## 1. Introduction

Gait is the periodic motion pattern of human walking or running. Different people owns different gait patterns, due to the reason that gait pattern is uniquely decided by the personal factors, such as personal habits, injury, and disease. Base on this character, researchers in pattern recognition area employ gait pattern to recognition the identity of walkers, namely gait recognition. And gait pattern is also used for disease diagnosing by the researchers in the field of medicine, namely gait analysis. No matter gait recognition or gait analysis, gait events detection is the basic problem of the both applications. Automatic detection of gait events is desirable for artificial intelligence applications, such as gait recognition and medicine abnormal gait analysis [1].

A gait cycle is the minimum periodic movement of human walking. Usually, a gait cycle is defined as a period from a heel strikes on the ground to the same heel strikes on the ground again the next time. According to the swing character of legs, a gait cycle can be divided into two phases, which are stance phase and swing phase. And there are also important six gait events within each gait cycle (shown as Figure 1), which are right heel strike, left toe-off, mid stance, left heel strike, right toe-off and mid swing. Accurate detection of the six gait events would raises the accuracy of gait recognition and analysis. In this paper, we focus on automatic detection of toe-off events using vision methods.

Currently, gait events detection methods can be mainly classified into two types: wearable sensors-based and vision-based methods [2]. The wearable sensors-based methods can accurately detect gait events by collecting motion data from the joints and segments of human lower limb with wearable devices. This type of method is widely used in the medicine area for evaluating abnormal gait due to its high accuracy performance. However, wearable sensors-based methods rely on high cooperation of participants. The participants have to first wear particular devices and then walk around the given area.

Conversely, vision-based methods detect gait event directly from video data captured by a single or several cameras without the aid of any other special sensors. Various cameras including structured light camera [3], stereo camera [4] and 2D vision camera [5] have been applied within these methods. Compared with the wearable sensors, cameras would be cheaper and easier to use. Detecting gait events from 2D video data is a challenging problem due to variations of illumination, perspective, and clothing. Previously, researchers attached markers to the joints of the human limb as participants walked on a clearly marked walkway. This setup requires the cooperation from participants.

In this paper, a new method of toe-off events detection based on a single 2D vision camera system is proposed. Consecutive pedestrian silhouettes extracted from video frames are combined to generate consecutive silhouettes difference maps (CSD-maps). Different number of consecutive silhouettes would result in different CSD-maps, namely *n*-CSD-maps, while *n* represents the number of consecutive silhouettes. Convolutional neural network is finally employed to learn the toe-off events detection features from CSD-maps. The main contribution of this paper is designing of a set of novel features, namely, consecutive silhouettes difference maps, for toe-off event detection. This method can be used to accurately detect gait event from video data captured from a single 2D vision camera under different viewing angles. If gait events can be accurately detected from 2D video data without participants cooperation, it would be greatly benefit to gait recognition and gait analysis.

The remainder of this study is organized as follows. In Section 2, the advancements of gait events detection methods are reviewed. In Section 3, the proposed method is discussed in detail. Section 4 reports the experimental results on publicly available databases. Finally, Section 5 concludes this study.

## 2. Related Work

In this section, we review the recent progress of gait event detection, which can be coarsely classified into two categories: wearable sensors-based methods and vision-based methods.

### 2.1. Wearable Sensors-Based Methods

Wearable sensors-based methods employ various wearable sensors placed on joints or segments of human limbs (such as feet, knees, thighs or waist) to collect their motion data. Accelerometers and gyroscopes are desirable sensors for gait event detection, which have drawn much attention from researchers. Rueterbories et al. [6] placed accelerometers on the foot to detect gait events. Aung et al. [7] placed tri-axial accelerometers on the foot, ankle, shank or waist to detect heel strike and toe off events. Formento et al. [8] placed a gyroscope on the shank to determine initial contact and foot-off events. Mannini et al. [9] used a uniaxial gyroscope to measure the angular velocity of foot instep in a sagittal plane. Anoop et al. [10] utilized force myography signals from thighs to determine the heel strike (HS) and toe-off (TO) events. Jiang et al. [11] proposed a gait phase detection method based on force myography technique.

The inertial measurement unit (IMU), which is composed of gyroscope and accelerometer, is also a powerful sensor for capturing human limb motion data. Bejarano et al. [12] employed two inertial and magnetic sensors placed on the shanks to detect gait events. Olsen et al. [13] accurately and precisely detected gait events using the features from trunk- and distal limb-mounted IMUs. And latter, Trojaniello et al. [14] mounted a single IMU at the waist level to detect gait events. Ledoux [15] presented a method for walking gait event detection using a single inertial measurement unit (IMU) mounted on the shank.

These sensors can accurately capture motion signals of the points where sensors are placed. Thus, these methods can accurately detect gait events and have been widely used for gait analysis in the medicine area. The disadvantages of these type of methods mainly lie in power consumption restriction, high cost and user cooperation restriction.

A smartphone would contain a 3-dimensional accelerometer, a 3-dimensional gyroscope, and a digital compass. Thus, smartphones are new convenient sensors for gait analysis. Pepa et al. [16] utilized smartphones to detection gait events (such as heel strike) by securing them to an individual’s lower back or sternum. Manor et al. [17] proposed a method to detect the heel strike and toe off events by placing a smartphone in the user’s pants pocket. Ellis et al. [18] presented a smartphone-based mobile application to quantify gait variability for Parkinson’s disease diagnosing. Smartphones are also powerful sensors for gait recognition. Fernandez-Lopez et al. [19] compared the performance of four state-of-art algorithms on a smartphone before 2016. Muaaz et al. [20] evaluated the security strength of a smartphone-based gait recognition system against zero-effort and live minimal-effort impersonation attacks under realistic scenarios. Gadaleta et al. [21] proposed a user authentication framework from smartphone-acquired motion signals. The goal of this work is to recognize a target user from their way of walking, using the accelerometer and gyroscope (inertial) signals provided by a commercial smartphone worn in the front pocket of the user’s trousers.

### 2.2. Vision-Based Methods

Vision-based methods can be also divided into two sub-categories: marker-based and no marker-based methods.

Marker-based methods calculate human limb motion parameters by tracking the markers attached to the joints of human limb. Ugbolue et al. [22] employed an augmented-video-based-portable-system (AVPS) to achieve gait analysis. In this study, bull’s eye markers and retroreflective markers are attached to human lower limb. In [23], Yang et al. proposed an alternative, inexpensive, and portable gait kinematics analysis system using a single 2D vision camera. Markers are also attached on the hip, knee, and ankle joints for motion data capture. And three years later, the authors enhanced the initial single-camera system by designing a novel autonomous gait event detection method [5]. These methods achieve good accuracy of gait event detection. However, a calibration step is needed, where the participant has to walk on a clearly marked walkway, thus indicating user cooperation is required.

No marker-based methods can achieve gait event detection without user cooperation. With respect to this type of method very few research studies have worked on gait event detection techniques. The directly related work is Auvinet’s work [3], in which a depth camera (Kinect) is employed to achieve gait analysis on a treadmill for routine outpatient clinics. In [3], a heel-strike event detection algorithm is presented by searching for extreme values of the distance between knee joints along the walking longitudinal axis. Although it achieves accurate detection results, Kinect used in [3] is also a special camera compared with a widely used web camera. In this study, we attempt to detect toe-off events using a web camera. As far as we know, this paper would be the first effort to detect gait events utilizing video data without the cooperation of participants.

Some research works about gait cycle detection algorithm have been presented in gait recognition methods. These methods can detect whole gait cycle or gait phase from video data without the help of markers. In [24], a gait periodicity detection method is presented based on dual-ellipse fitting (DEF). The periodicity is defined as the internal between the first extreme point and the third extreme point of DEF signals. Kale et al. [25] employed the norm of the width vector to show a periodic variation. Sarkar et al. [26] estimated gait cycle by counting the number of foreground pixels in the silhouette in each frame overtime. Mori et al. [27] detected the gait period by maximizing the normalized autocorrelation of the gait silhouette sequence for the temporal axis. These methods mentioned above can achieve gait cycle detection, but cannot obtain accurate gait event detection results.

## 3. Toe-Off Events Detection Based on CSD-Maps

In this section, we present the technique detail of the toe-off events detection method. The framework of the proposed method is graphically presented in Figure 2. Several consecutive silhouettes of a pedestrian are first combined to generate a consecutive silhouettes difference map. Convolutional neural networks are then employed to learn the features for toe-off events classification.

### 3.1. Consecutive Silhouettes Difference Maps

There are rich temporal and spatial information contained in video data. Mining and fusing temporal and spatial information is currently an interest in computer vision. Inspired by the principle of the exclusive OR operation, we employ a frame difference method to encode the temporal and spatial information contained in several consecutive frames into a map. The difference map generated from *n* consecutive silhouettes is named as a *n*-CSD-map. We first take a 2-CSD-map as an example to explain how consecutive silhouette frames are encoded into a map.

#### 3.1.1. 2-CSD-Maps

The main idea of 2-CSD-maps is graphically presented in Figure 3. The 2-CSD-map of the $i^{th}$ frame is generated from two consecutive silhouette frames. Let $\Gamma_{i}^{2}$ present the 2-CSD-map of the $i^{th}$ frame, $I_{i-1}$ and $I_{i}$ present the binary silhouette images of the ${\left(i-1\right)}^{th}$ and $i^{th}$ frame. For any pixel $P_{j,k}^{2}$ in $\Gamma_{i}^{2}$, it’s pixel value can be formulated as following:(1)$$\Gamma_{i}^{2}\left(j,k\right)=\left\{\begin{array}{cc} 1 & \mathrm{if}\;\left(P_{j,k}^{2}\notin \Omega_{i-1}\right)\cap \left(P_{j,k}\in \Omega_{i}\right)\\ 2 & \mathrm{if}\;\left(P_{j,k}^{2}\in \Omega_{i-1}\right)\cap \left(P_{j,k}\notin \Omega_{i}\right)\\ 3 & \mathrm{if}\;\left(P_{j,k}^{2}\in \Omega_{i-1}\right)\cap \left(P_{j,k}\in \Omega_{i}\right) \end{array}\right.$$
while, $\Omega_{i-1}$ represents the pixel set of the silhouette area in $I_{i-1}$, and $\Omega_{i}$ represents the pixel set of the silhouette in $I_{i}$. In order to achieve a good visual effect, the pixel values in Figure 3c are normalized to [0,1].

In practice, a pedestrian silhouette is presented as a binary image. Thus, a 2-CSD-map of two consecutive silhouettes can be computed using following three steps to achieve fast extraction of 2-CSD-maps.

First, copy gray value of pixels from $I_{i-1}$ to $\Gamma_{i}^{2}$. A temporary matrix *I* is then computed as:(2)$$I=I_{i}-I_{i-1}$$

Secondly, modify the pixel value of $\Gamma_{i}^{2}$ according to the value of matrix *I*:(3)$$\Gamma_{i}^{2}\left(j,k\right)=\left\{\begin{array}{cc} 1 & \mathrm{if}\;I(j,k)>0\\ 2 & \mathrm{if}\;I(j,k)<0 \end{array}\right.$$

Finally, the pixel value of $\Gamma_{i}^{2}$ is modified as follows:(4)$$\Gamma_{i}^{2}\left(j,k\right)=\left\{\begin{array}{cc} 3 & \mathrm{if}\;\Gamma_{i}^{2}\left(j,k\right)==255\\ \Gamma_{i}^{2}\left(j,k\right) & \mathrm{else} \end{array}\right.$$

Some samples of 2-CSD-maps are graphically presented in Figure 4. We can see that 2-CSD-maps are distinctive features for toe-off events detection compared with original silhouette images.

#### 3.1.2. *n*-CSD-Maps

Suppose that there are *n* consecutive silhouettes images $I_{1}$, $I_{2}$, ..., and $I_{n}$. The *n*-CSD-maps $\Gamma_{i}^{n}$ can be formulated as following:(5)$$\Gamma_{i}^{n}\left(j,k\right)=\left\{\begin{array}{cc} 1 & \mathrm{if}\;\left(P_{j,k}^{n}\in \Omega_{1}\right)\cap \left(P_{j,k}\notin \Omega_{2}\right)\cap \left(P_{j,k}\notin \Omega_{3}\right)\cap ...\cap \left(P_{j,k}\notin \Omega_{n}\right)\\ 2 & \mathrm{if}\;\left(P_{j,k}^{n}\notin \Omega_{1}\right)\cap \left(P_{j,k}\in \Omega_{2}\right)\cap \left(P_{j,k}\notin \Omega_{3}\right)\cap ...\cap \left(P_{j,k}\notin \Omega_{n}\right)\\ 3 & \mathrm{if}\;\left(P_{j,k}^{n}\notin \Omega_{1}\right)\cap \left(P_{j,k}\notin \Omega_{2}\right)\cap \left(P_{j,k}\in \Omega_{3}\right)\cap ...\cap \left(P_{j,k}\notin \Omega_{n}\right)\\ ....\\ 2^{n}-1 & \mathrm{if}\;\left(P_{j,k}^{n}\in \Omega_{1}\right)\cap \left(P_{j,k}\in \Omega_{2}\right)\cap \left(P_{j,k}\in \Omega_{3}\right)\cap ...\cap \left(P_{j,k}\in \Omega_{n}\right) \end{array}\right.$$
while, $\Gamma_{i}^{n}\left(j,k\right)$ stands for the pixel value of the pixel $P_{j,k}^{n}$ in the generated *n*-CSD-map. $\Omega_{1}$, $\Omega_{2}$, …, and $\Omega_{n}$ represent the pixel set of the silhouette areas in frame $I_{1}$, $I_{2}$, …, and $I_{n}$ respectively.

Given *n* consecutive silhouette images, the *n*-CSD-maps extraction algorithm can be described as Algorithm 1. With this algorithm, the CSD-map generated from the given consecutive silhouette images is also presented as an image with the same size as silhouette images, shown as Figure 3c. Thus, a further normalization step is necessary. In this paper, CSD-map images are initially normalized to a certain size (such as 90 × 140) using Algorithm 2.

Figure 5 shows some consecutive normalized CSD-maps. we can see that the CSD-maps under toe-off state are obviously different with other CSD-maps.

**Algorithm 1** Algorithm for generating *n*-CSD-maps**Require:** Consecutive silhouette images: I[w,h,n]. Parameter *w* and *h* represent the width and height of the silhouette images respectively. Parameter *n* represents the number of consecutive silhouette images.**Ensure:** The CSD-map: Γ1: **for**i=1 to *w*
**do**2:  **for**
j=1 to *h*
**do**3:   t=I(i,j,:);4:   value=0;5:   **for**
k=1 to *n*
**do**6:    value=value+2(k−1)∗t(k);7:   **end for**8:   Γ(i,j)=value;9:  **end for**10: **end for**11: return Γ;

**Algorithm 2** Algorithm for normalizing a CSD-map**Require:** The original CSD-map image: OM The width of the normalized CSD-map:*w* The height of the normalized CSD-map:*h***Ensure:** The normalized CSD-map: NM1: [x,y]=find(OM>0);2: segm=OM(min(x):max(x),min(y):max(y));3: NM=imresize(segm,[h,w]);4: return NM;

### 3.2. Convolutional Neural Network

Convolutional neural networks have a feed-forward network architecture with multiple interconnected layers which may be of any of the following types: convolution, normalization, pooling and fully connected layers. CNNs have recently achieved many successes in visual recognition tasks, including image classification [28], object detection [29], and scene parsing [30]. CNNs are chosen as a detector for this study because they outperform other traditional methods in many image classification challenges, such as ImageNet [28] and many other image-based recognition problems, e.g., face recognition and digital recognition [31]. Comparing with traditional methods which rely on feature engineering, CNNs are able to learn feature representation through the back propagation algorithm without the need for much intervention and also achieve much higher accuracy.

The aim of this study is not to propose another CNN but use a classic CNN to address the problem of toe-off events detection. In this paper, we employ the CNN architecture presented in Figure 6. It is modified from DeepID [32]. The network includes three convolutional layers and three fully connected layers. The three convolutional layers have 64, 128 and 256 kernels and their sizes are respectively 5 × 5, 3 × 3 × 64 and 3 × 3 × 128. The first fully connected layer has 1024 neurons and the second fully connected layer has 512 neurons. In the last fully connected layer, there are two neurons, one for toe-off frame output and the other for non-toe-off frame output. The max-pooling with a size of 2 and a stride of 2 follows the three convolutional layers.

## 4. Experiments and Results Analysis

### 4.1. Database

Experiments are conducted on CASIA gait database (Dataset B) [33] to evaluate the accuracy of the performance of the proposed method. The data contained in this database are collected from 124 subjects (93 males and 31 females) in an indoor environment under 11 different viewing angles. The data from a subject is simultaneously captured by 11 USB cameras (with a resolution of 320 × 240, and a frame rate of 25 fps) around the left hand side of the subject when he/she was walking, and the angle between two nearest view directions is 18${}^{\circ }$. When a subject walked in the scene, he/she was asked to walk naturally along a straight line 6 times first, and 11 × 6 = 66 normal walking video sequences were captured for each subject. After normal walk, the subjects were asked to put on their coats or carried a bag, and then walked twice along the straight line. In each viewing angle, there are totally 10 videos collected from every subject under three different clothing conditions, namely normal condition, coat condition and bag condition. The CASIA Gait Database is provided free of charge at web site http://www.cbsr.ia.ac.cn.

In this study, we considered the data captured under the viewing angles of 36${}^{\circ }$, 54${}^{\circ }$, 72${}^{\circ }$, 90${}^{\circ }$, 108${}^{\circ }$, 126${}^{\circ }$ and 144${}^{\circ }$ (approximately 500,000 frames in total) for training and testing. The data captured under the frontal viewing angles of 0${}^{\circ }$, 18${}^{\circ }$, 172${}^{\circ }$, 180${}^{\circ }$, are not used in the experiments primarily because there is very little difference between two consecutive silhouettes. The CSD-maps generated from the video data captured in the viewing of sagittal plane do not contain much information for gait events detection. This means that the method proposed by this paper cannot deal with the video data captured in the viewing of sagittal plane. Even so, the proposed method can deal with the video data captured from most viewing angles. This makes the proposed method useful in practice.

### 4.2. Toe-Off Frame Definition and Data Preparation

The ground truth of all the silhouette frames should be manually labeled for modal training and testing. Thus, the toe-off frames should be first and clearly defined.

Human gait is a continuous and periodic movement. In medical field, the toe-off event is defined as the moment that the stance limb leaves the ground, shown as in Figure 1. While, the video data is the sampling record of human gait with a certain frame rate θ. Usually, the frame rate θ would be 30 fps. And the gait cycle of a person is averagely about 1 s time consuming. This means that one gait movement cycle of a person would be recoded as about 30 consecutive frames with an interval of 33 ms. The problem is that the moment the stance limb leaving the ground may not be included in the 30 consecutive sampling frames. In this paper, the first frame after the stance limb leaves the ground is defined as a toe-off frame. For example, as shown in Figure 7, if the moment that the stance limb leaves the ground falls within the period of $t_{n}<t<t_{n+1}$, then the frame (*n* + 1) is defined as the toe-off frame.

According to the definition, there would exist error in the labeled groundtruth. Let θ be the frame rate of the video data. The during time between two continuous frames would be $\frac{1}{\theta }$, which means $t_{n+1}-t_{n}=\frac{1}{\theta }$. If the toe-off event happens during the period of $\left(t_{n},t_{n+1}\right)$ but nearer to $t_{n}$ shown as Figure 7a, then at frame *n* + 1, the foot would have swung in the air for about $\frac{1}{\theta }$ seconds. However, if the toe-off event happens during the period of $\left(t_{n},t_{n+1}\right)$ but nearer to $t_{n+1}$ shown as Figure 7b. At frame *n* + 1, the foot would have just left the ground. The frames *n* + 1 in both Figure 7a,b are regarded as toe-off frames. Obviously, the toe-off frames in Figure 7a,b may be different with each other. But this error doesn’t change the validity of the proposed method.

### 4.3. Experimental Configuration

The experiments are conducted by using Caffe [34], which is a deep learning framework created by Yangqing Jia during his PhD at UC Berkeley. The experiments are conducted as following.

*Configuration of n-CSD-maps.* Several pre-tests have been conducted under the viewing angles of 72${}^{\circ }$, 90${}^{\circ }$ and 108${}^{\circ }$ for choosing the size of normalized CSD-maps. As shown in Figure 8, the pre-test results show that different sizes of normalized CSD-maps practically cause almost no change to the detection accuracy. The main reason is that CSD-maps are generated from binary pedestrian silhouettes. The decline of the size of normalized CSD-maps would not result in much change to the detection accuracy of this method. Thus, in the following experiments, the size of normalized CSD-maps is set as 48*32. As to the parameter *n* of *n*-CSD-maps, it is set as 2, 3, 4, 5, and 6. This means that 2-CSD-maps, 3-CSD-maps, 4-CSD-maps, 5-CSD-maps and 6-CSD-maps are used in the experiments. The reason is that the increase of parameter *n* brings little increase of detection accuracy, while costs more time for features extraction, shown as Table 1 and Figure 9.*Configuration of Training set and test set.* The samples from subject #001 to subject #90 of each viewing angle are selected for model training. The rest of samples (from subject #091 to subject #124) is used for testing.*Configuration of CNN Solver.* The initialized learning rate is 0.001, the momentum is 0.9 and the weight decay is 0.0005. The maximum number of iteration in each experiment is 20,000. The weights in the CNN are initialized with a zeromean Gaussian distribution with standard deviation of 0.0001. The bias is set to one.

### 4.4. Experimental Results and Discussion

In this paper, a new evaluation indicator, namely n-frame-error cumulative detection accuracy, is designed to evaluate the performance of the proposed method besides detection accuracy and ROC curve. The n-frame-error cumulative detection accuracy is similar with cumulative match characteristics (CMC) curves [35]. Let’s *d* represents the difference between the sequence number of predicted toe-off frame and the ground truth, shown as Figure 10. *n*-frame-error cumulative detection accuracy indicates the detection accuracy with the condition of d≤n.

Table 1 shows the detection accuracy of the proposed method. We can see that the proposed method achieves good detection accuracy. The proposed method reaches the accuracy around 93% under the viewing angles of 36${}^{\circ }$, and achieves the peak value of 93.63% by using 6-CSD-maps. Under the viewing angle of 54${}^{\circ }$, the proposed method reaches the accuracy around 94% and achieves the peak value of 95.4% by using 6-CSD-maps. Under the viewing angle of 72${}^{\circ }$, the proposed method reaches the accuracy around 95% and achieves the peak value of 95.44% by using 6-CSD-maps. Under the viewing angle of 90${}^{\circ }$, the proposed method reaches the accuracy around 96% and achieves the peak value of 96.78% by using 6-CSD-maps. Under the viewing angle of 108${}^{\circ }$, the proposed method reaches the accuracy around 95% and achieves the peak value of 95.78% by using 6-CSD-maps. Under the viewing angle of 126${}^{\circ }$, the proposed method also reaches the accuracy around 95% and achieves the peak value of 95.65% by using 6-CSD-maps. Under the viewing angle of 144${}^{\circ }$, the proposed method reaches the accuracy around 93% and achieves the peak value of 93.44% by using 6-CSD-maps.

The relationship between detection accuracy of the proposed method and *n*-CSD-map is graphically presented in Figure 9. Figure 9a demonstrates the detection accuracy of the proposed method as a function of *n*-CSD-map, and the corresponding bars are presented in Figure 9b. Generally, the detection accuracy is slightly improved with the increase of *n*. The reason is that the bigger the parameter *n* is, the more consecutive silhouettes will be encoded into a CSD-map, and the more information will be contained in the CSD-map. The detection accuracy gets a good promotion when the parameter *n* changes from 2 to 3. For example, under viewing angle of 108${}^{\circ }$, the accuracy of the proposed method increase from 94.68% to 95.54% when the parameter *n* increases from 2 to 3. However, the accuracy gets a few increase when the parameter *n* goes to 4, 5 and 6. This demonstrates that 3-CSD-map is a good choice for toe-off detection, which can achieve good accuracy with little additional computation cost. Figure 11 shows the ROC curves of the proposed method under different viewing angles. The ROC curves of the proposed method under the viewing angles of 36${}^{\circ }$, 54${}^{\circ }$, 72${}^{\circ }$, 90${}^{\circ }$, 108${}^{\circ }$, 126${}^{\circ }$ and 144${}^{\circ }$ are respectively presented in the figures from Figure 11a–g. As shown in Figure 11, under all viewing angles, the proposed method gets higher detection performance by using larger parameter *n* of *n*-CSD-map.

The ROC curves of the proposed method using 3-CSD-map under different viewing angles are presented in Figure 9h. Generally, we can see that the proposed method obtains higher detection accuracy around coronal plane viewing angles than sagittal plane viewing angles. Especially, the proposed method achieves the accuracy of 96.78% under the viewing angle 90${}^{\circ }$, which is higher than other viewing angles. This demonstrate that CSD-maps generated from the video data captured in sagittal plane viewing angles contain less useful information for gait events detection than coronal plane viewing angles. The reason is that there is fewer different between two consecutive silhouettes of video frames captured under sagittal plane viewing angles compared with coronal plane viewing angles.

The plots presented in Figure 12a are the *n*-frame-errors cumulative detection accuracy of the proposed method against different viewing angles. The 1-frame-error cumulative detection accuracy of the proposed method reaches the accuracy of 99.3%, 99.86%, 99.9%, 99.9%, 99.9%, 99.8%, and 99.4% for the viewing angles of 36${}^{\circ }$, 54${}^{\circ }$, 72${}^{\circ }$, 90${}^{\circ }$, 108${}^{\circ }$, 126${}^{\circ }$, and 144${}^{\circ }$ respectively. For the 2-frame-error, the cumulative detection accuracy of the proposed method achieves 100% for the viewing angles of 54${}^{\circ }$, 72${}^{\circ }$, 90${}^{\circ }$, 108${}^{\circ }$, and 126${}^{\circ }$. This demonstrates that the maximum time error of the proposed method detecting toe-off events in coronal plane viewing angles is less than $\frac{2}{\theta }$, where θ is the frame rate of the video data. Practically, we can promote the time accuracy of this method by increasing the frame rate of the video.

Figure 12b shows the detection accuracy of the proposed method as a function of viewing angles compared with [24,25,36]. Due to the reason that [24,25] do not provide toe-off event detection results directly, we implemented the both algorithms for toe-off event detection according to the main ideas of [24,25]. Ref. [36] is our previous work based on principal component analysis and support vector machine. In this experiment, all frames are used for training and testing in 5-fold cross validation. We can see that our CNN-based method significantly outperforms Ben’s method [24], Kale’s method [25] and our previous work [36] in the viewing angles of 36${}^{\circ }$, 54${}^{\circ }$, 72${}^{\circ }$, 90${}^{\circ }$, 108${}^{\circ }$, 126${}^{\circ }$, and 144${}^{\circ }$.

In Figure 13, we use a confusion matrix to evaluate cross viewing angle detection accuracy of this method using 3-CSD-maps. As can be seen in the figure, this method achieves the best accuracy in the counter-diagonal and around 90% in the other areas, which means that this method can get good accuracy for cross view toe-off detection. Figure 14 presents the ROC curves of this method under all viewing angles compared with [24,25,36]. We can see that the proposed method significantly outperforms the comparation methods.

## 5. Conclusions and Future Work

This paper presents a promising vision-based method to detect toe-off events. The main contribution of this paper is the design of consecutive silhouettes difference maps for toe-off event detection. Convolutional neural network is employed for feature dimension reduction and toe-off event classification. Experiments on a public database have demonstrated good performance of our method in terms of detection accuracy. The main advantages of the proposed method can be described as following.

Comparing with wearable sensors-based methods, this method can detect toe-off event from 2D video data without the cooperation of participants. Usually, in the field of medicine, wearable sensors-based methods are the first choice for gait analysis, due to their high accuracy. However, these methods are suffering the disadvantages of high cooperation from users and power consumption restriction. The method proposed by this paper, which also achieves good accuracy for toe-off event detection by using a web camera, can overcome the disadvantages of wearable sensors-based methods for gait analysis.Comparing with other vision-based methods, this method provides a better accuracy for toe-off event detection. Gait cycle detection is a basic step of gait recognition. An accurate toe-off event detection algorithm can produce an accurate gait cycle detection algorithm. Thus, the method proposed by this paper would be beneficial to gait recognition.

Although a promising feature representation method is proposed in this paper for toe-off event detection, more efforts are needed to improve the method of gait events detection from video data in our future work.

A much larger database is needed to test the practical performance of toe-off event detection under different conditions.CSD-map provides a good feature representation for detecting toe-off events from video data. It also would be applicable for other gait events detection, such as heel strike, foot flat, mid-stance, heel-off, and mid-swing.

## Figures and Tables

**Figure 1 entropy-21-00329-f001:**
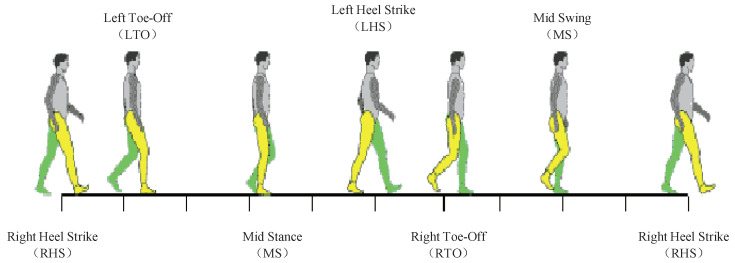
Graphic demonstration of the gait events within a gait cycle.

**Figure 2 entropy-21-00329-f002:**
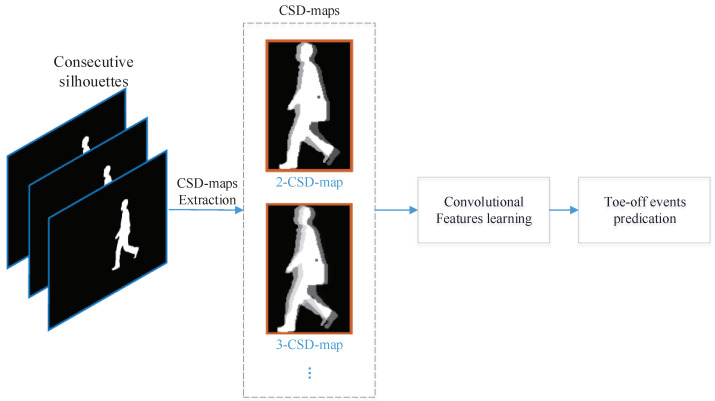
The framework of the proposed method.

**Figure 3 entropy-21-00329-f003:**
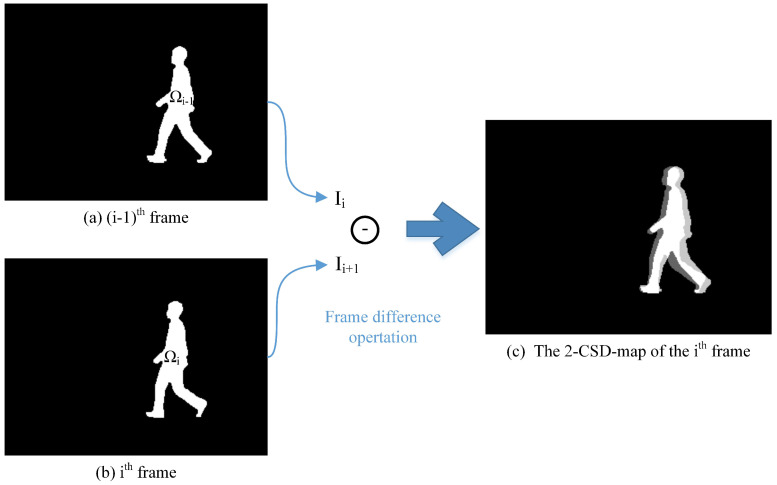
The basic idea of the 2-CSD-map. The pixel values in (**c**) are are normalized to [0,1] for good visual effect.

**Figure 4 entropy-21-00329-f004:**
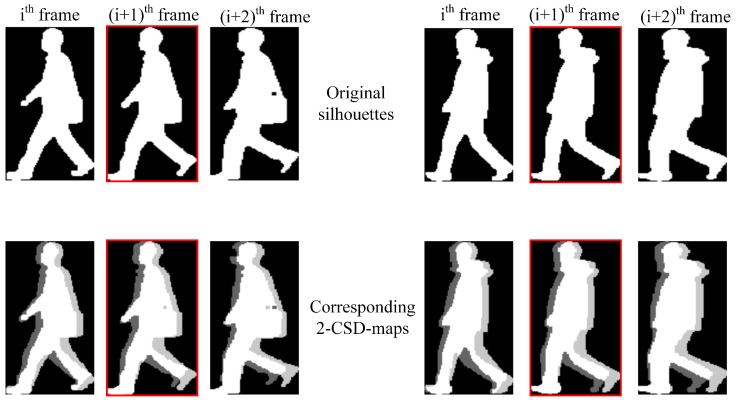
Samples of 2-CSD-maps compared with original silhouettes. The images presented in the first row are original silhouettes of two different persons, and the corresponding 2-CSD-maps are presented in the second row. The images with red edging are the toe-off frames. The pixel values in 2-CSD-maps are normalized to [0,1] for good visual effect.

**Figure 5 entropy-21-00329-f005:**
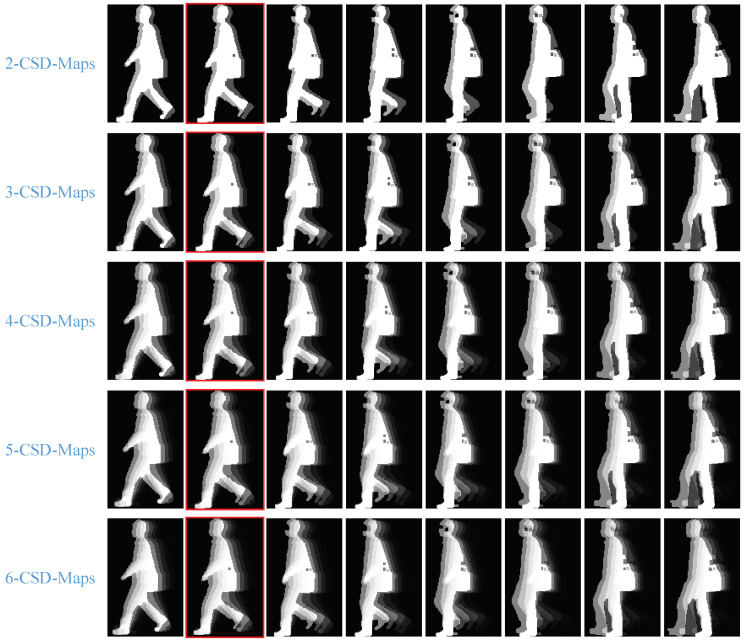
Samples of normalized CSD-maps. From the first row to the fifth row, the normalized 2-CSD-maps, 3-CSD-maps, 4-CSD-maps, 5-CSD-maps and 6-CSD-maps are respectively presented. The images with red edging are the toe-off frames. The pixel values in all CSD-maps are normalized to [0,1] for good visual effect.

**Figure 6 entropy-21-00329-f006:**
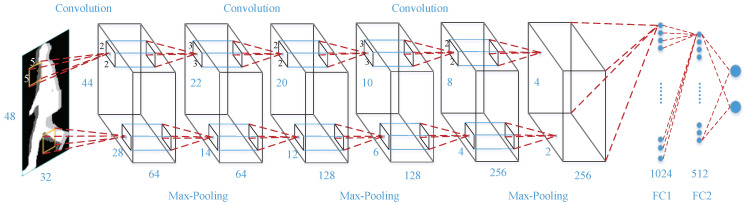
The architecture of the CNN employed in this study.

**Figure 7 entropy-21-00329-f007:**
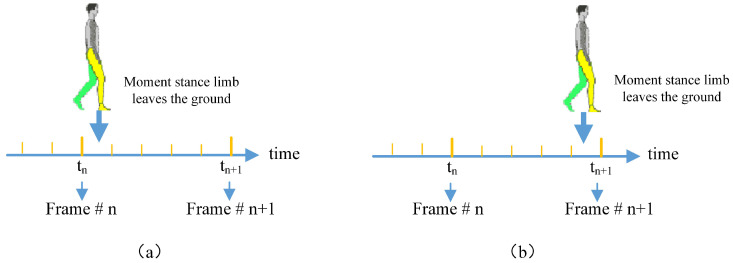
Toe-off event definition of video frames.

**Figure 8 entropy-21-00329-f008:**
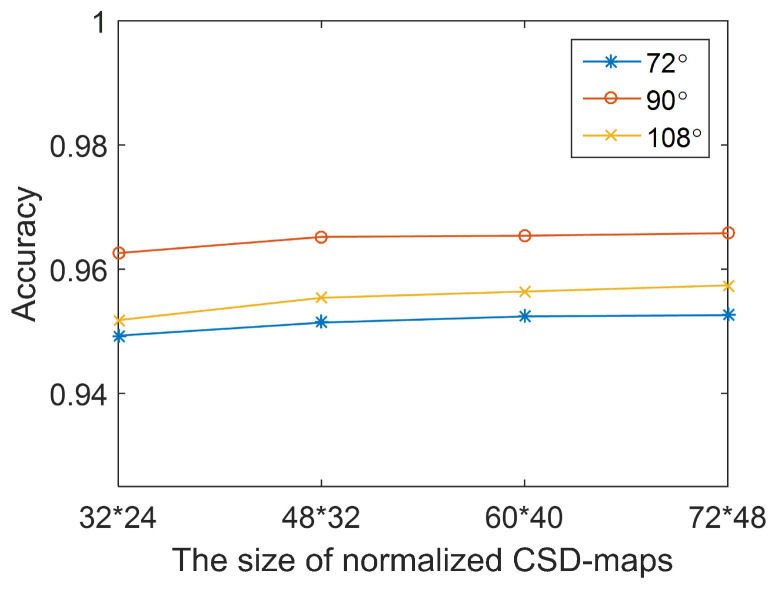
The relationship between detection accuracy of the proposed method and the size of normalized *n*-CSD-map.

**Figure 9 entropy-21-00329-f009:**
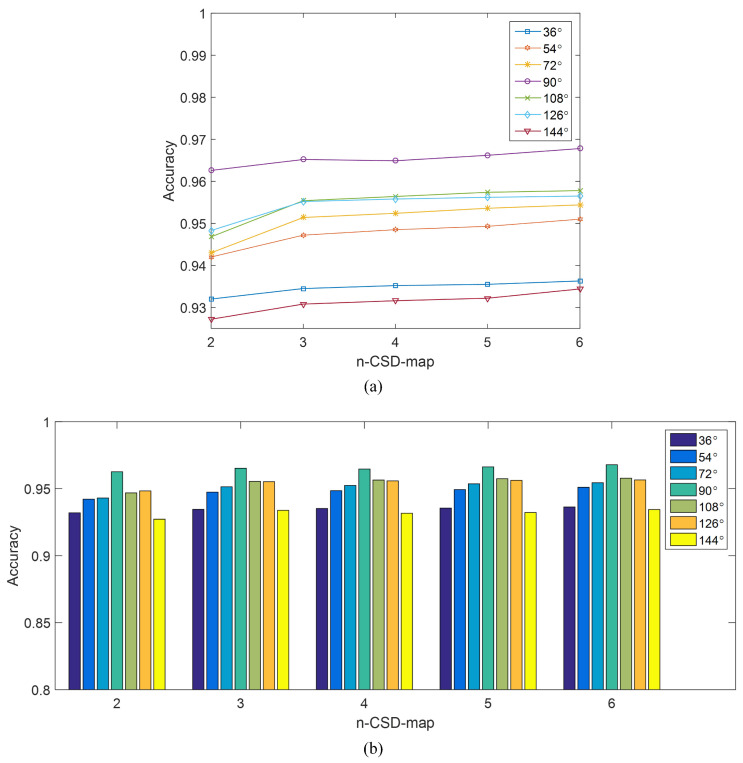
The relationship between detection accuracy of the proposed method and *n*-CSD-map. (**a**) The detection accuracy as a function of *n*-CSD-map. (**b**) The bars of the detection accuracy VS. *n*-CSD-map.

**Figure 10 entropy-21-00329-f010:**
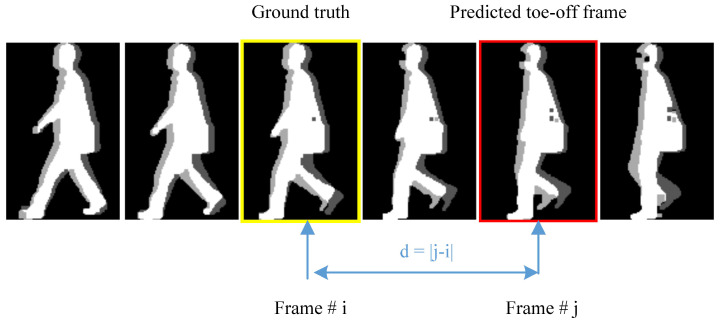
Graphical demonstration of the n-frame-error cumulative detection accuracy. The frame difference between the sequence number of predicted toe-off frame and the ground truth is noted as *d*. The image with red edging is the predicted toe-off frame. The image with yellow edging is the ground truth.

**Figure 11 entropy-21-00329-f011:**
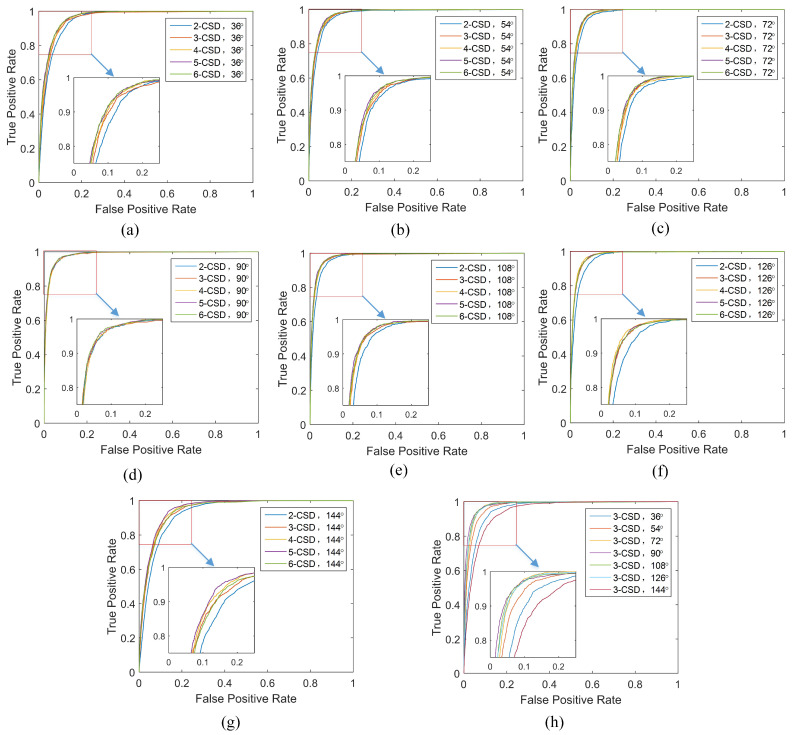
The ROC curves of the proposed method. (**a**) The ROC curves under the viewing angle of 36${}^{\circ }$. (**b**) The ROC curves under the viewing angle of 54${}^{\circ }$. (**c**) The ROC curves under the viewing angle of 72${}^{\circ }$. (**d**) The ROC curves under the viewing angle of 90${}^{\circ }$. (**e**) The ROC curves under the viewing angle of 108${}^{\circ }$. (**f**) The ROC curves under the viewing angle of 126${}^{\circ }$. (**g**) The ROC curves under the viewing angle of 144${}^{\circ }$. (**h**) The ROC curves of the proposed method with 3-CSD-map under different viewing angles of 36${}^{\circ }$, 54${}^{\circ }$, 72${}^{\circ }$, 90${}^{\circ }$, 108${}^{\circ }$, 126${}^{\circ }$ and 144${}^{\circ }$.

**Figure 12 entropy-21-00329-f012:**
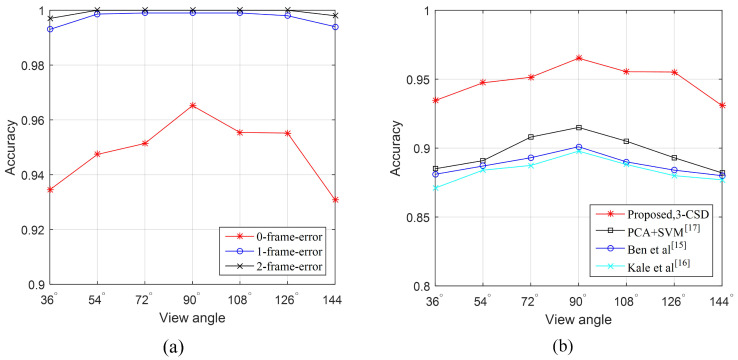
The n-frame-error cumulative detection accuracy of the proposed method. (**a**) the detection accuracy of the proposed method against different frame-errors. (**b**) The detection accuracy of the proposed method compared with [24,25,36].

**Figure 13 entropy-21-00329-f013:**
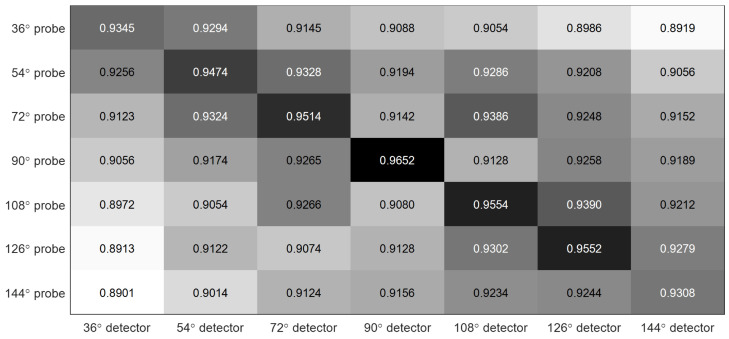
The confusion matrix of cross viewing angle detection accuracy of this method using 3-CSD-maps.

**Figure 14 entropy-21-00329-f014:**
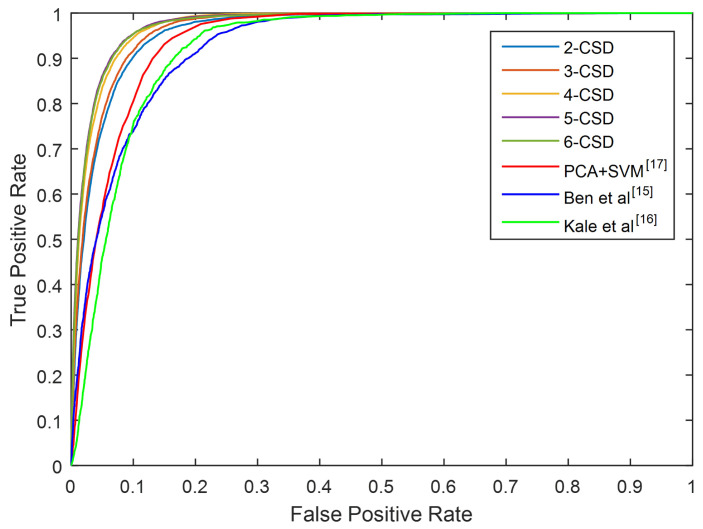
The ROC curves of this method compared with [24,25,36] under all viewing angles.

**Table 1 entropy-21-00329-t001:** Detection accuracy of the proposed method.

*n*-CSD-Maps	36 Degree	54 Degree	72 Degree	90 Degree	108 Degree	126 Degree	144 Degree
2-CSD	93.2%	94.34%	94.3%	96.26%	94.68%	94.83%	92.72%
3-CSD	93.45%	94.74%	95.14%	96.52%	95.54%	95.52%	93.08%
4-CSD	93.52%	95.18%	95.24%	96.58%	95.64%	95.58%	93.16%
5-CSD	93.55%	95.38%	95.36%	96.62%	95.74%	95.62%	93.22%
6-CSD	**93.63%**	**95.4%**	**95.44%**	**96.78%**	**95.78%**	**95.65**%	**93.44%**

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
