# Peer review of "Detecting Toe-Off Events Utilizing a Vision-Based Method"

_entropy, 2019, doi:10.3390/e21040329_

Round 1

Reviewer 1 Report

The authors designed a set of features using consecutive silhouettes difference maps for toe-off event detection. They used a vanilla CNN for testing the validity of their vision based method. The performance reported probes that this method could be useful, even though a more in-depth comparison study will be required. However, the simplicity and novelty of the idea are worthy of being published. 

As for suggestions to improve the paper, the hyperparameters should have been tuned, and the experimental configuration might be explained in more detail, particularly the" configuration of n-CSD maps" process.

Finally, equation 2 is confusing because F_i is defined in the body of the text instead than in an equation. 

Author Response

Response to Reviewer 1 Comments

Point 1: As for suggestions to improve the paper, the hyperparameters should have been tuned, and the experimental configuration might be explained in more detail, particularly the "configuration of n-CSD maps" process.

Response 1: Thanks for your suggestion! The experimental configuration of this paper is described (from line 219 to line 228, and figure 8) in more detail.

Point 2: Finally, equation 2 is confusing because F_i is defined in the body of the text instead than in an equation.

Response 2: Thanks for your suggestion! The symbols of “F_i” in the equations of 1, 2, 3, 4, and 5 are revised.

Reviewer 2 Report

In introduction and related work maybe options based on smartphone sensors maybe include.

It could be interesting if in conclusions indicate potential benefit of this approach against others.   

More information about datasets could be useful to other researchers interested in replication.

Author Response

Response to Reviewer 2 Comments

Point 1: In introduction and related work maybe options based on smartphone sensors maybe include.

Response 1: Thanks for your suggestion! A paragraph (from line 82 to line 94) is added to describe smartphone sensors based gait analysis and recognition methods.

Point 2: It could be interesting if in conclusions indicate potential benefit of this approach against others.

Response 2: Thanks for your suggestion! Several sentences (lines from 307 to 319) are added to describe the potential benefit of this approach against others.

Point 3: More information about datasets could be useful to other researchers interested in replication.

Response 3: Thanks for your suggestion! More information (from line 177 to line 187) about CASIA gait database (Dataset B) is added to this version.
